# Isolation of Electrochemically Active Bacteria from an Anaerobic Digester Treating Food Waste and Their Characterization

**DOI:** 10.3390/microorganisms12081645

**Published:** 2024-08-11

**Authors:** Daichi Yoshizu, Soranosuke Shimizu, Miyu Tsuchiya, Keisuke Tomita, Atsushi Kouzuma, Kazuya Watanabe

**Affiliations:** 1Laboratory of Bioenergy Science and Technology, School of Life Sciences, Tokyo University of Pharmacy and Life Sciences, Tokyo 192-0392, Japan; 442daichi@yahoo.ne.jp (D.Y.); s208032@toyaku.ac.jp (S.S.); uktomita@toyaku.ac.jp (K.T.); akouzuma@toyaku.ac.jp (A.K.); 2J &T Recycling Corporation, Yokohama 230-0044, Japan; tsuchiya-miyu@jt-kankyo.co.jp

**Keywords:** bioelectrochemistry, anaerobic digestion, exoelectrogen, electroactive microbe, extracellular electron transfer

## Abstract

Studies have used anaerobic-digester sludge and/or effluent as inocula for bioelectrochemical systems (BESs), such as microbial fuel cells (MFCs), for power generation, while limited studies have isolated and characterized electrochemically active bacteria (EAB) that inhabit anaerobic digesters. In the present work, single-chamber MFCs were operated using the anaerobic-digester effluent as the sole source of organics and microbes, and attempts were made to isolate EAB from anode biofilms in MFCs by repeated anaerobic cultivations on agar plates. Red colonies were selected from those grown on the agar plates, resulting in the isolation of three phylogenetically diverse strains affiliated with the phyla *Bacillota*, *Campylobacterota* and *Deferribacterota*. All these strains are capable of current generation in pure-culture BESs, while they exhibit different electrochemical properties as assessed by cyclic voltammetry. The analyses of their cell-free extracts show that cytochromes are abundantly present in their cells, suggesting their involvement in current generation. The results suggest that anaerobic digesters harbor diverse EAB, and it would be of interest to examine their ecological niches in anaerobic digestion.

## 1. Introduction

Bioelectrochemical systems (BESs) are devices that examine and/or utilize the electrochemical properties of biological agents, such as microbes and enzymes [1]. Studies have developed a variety of microbial BESs (MESs), including microbial fuel cells (MFCs) and microbial electrolysis cells (MECs), in which electrochemically active bacteria (EAB) able to perform extracellular electron transfer (EET) play pivotal roles [2]. Among these, MFCs generate electric power at the expense of organic matter [3], while MECs produce hydrogen and/or methane gases from organic matter with the aid of electric power [4]. These devices are expected to be applied to the development of sustainable societies, since energy can be generated from biomass waste and wastewater discharged from human activities [2].

In order to initiate the operation of MESs, researchers have inoculated MESs with microbe-bearing natural samples, such as wastewater, soils, sediments, aerobic sludge and anaerobic sludge. Heidrich et al. used the microbes in soil and wastewater for power generation in MFCs, suggesting that the microbial diversity in inoculum does not impact the power generation [5]. Another study used aerobic or anaerobic sludge as inoculum to start up air-cathode single-chamber MFCs, suggesting that aerobic sludge performed better for the MFC startup than anaerobic sludge, while power outputs ultimately converge after the establishment of anode communities [6]. In addition, studies have demonstrated that the inoculation with anaerobic-digester sludge facilitates power generation in MFCs [7,8]. Likewise, many studies have used sludge and effluents taken from anaerobic digesters as inocula for MFCs and MECs, suggesting that EAB are commonly present in anaerobic digesters.

So far, a variety of EAB have been identified and/or isolated from environmental samples, and these are affiliated with diverse taxonomic groups [9]. The identification of EAB would be possible in several different ways, including isolation coupled to pure-culture examinations [10,11] and meta-omics analyses [12,13,14]. For instance, many EAB affiliated with *Shewanella* and *Geobacter* spp. have been isolated from marine and freshwater sediments, and their electrochemical activities have been examined [15,16,17]. In addition, metatranscriptomic approaches have been used to identify EAB occurring in MFCs treating wastewater [12]. On the other hand, although metabarcoding analyses have detected *Geobacter* relatives from anode biofilms in MFCs inoculated with anaerobic sludge [18,19], to our knowledge, no study has isolated EAB from anaerobic digesters.

In the present work, in order to deepen our understanding of EAB inhabiting anaerobic digesters, we attempted to isolate EAB from effluents discharged from an anaerobic digester treating food wastes. It is shown that EAB isolated from the anaerobic digester are more diverse than previously thought, suggesting that anaerobic digesters are habitats for diverse EAB.

## 2. Materials and Methods

### 2.1. Enrichment of EAB

The anaerobic-digester effluent (ADE) used in this work was obtained from an anaerobic digester treating food wastes. The chemical oxygen demand (Cr) concentration (COD) was approximately 40,000 mg L^−1^. The MFC used in this study for the enrichment of EAB on anode surfaces were described previously [20]. Briefly, the MFC was a single-chamber reactor [3], 85 mL in capacity, and equipped with a graphite-felt anode (20 cm^2^) and an air cathode (25 cm^2^). MFCs were filled with ADE (80 mL), which served as the electrolyte, organic substrate and the sole source of microbes. The mixture was gently agitated using a magnetic stirrer at approximately 100 rpm. The anode and cathode were connected via an external resistor (*R*, Ω; initially 2000 Ω), and the voltage across the resistor (*E*, V) was measured using a data logger (GL820, Graphtec, Yokohama, Japan). In this work, this resistor was chosen based on the results of the previous work [20]. A current density (*J*, mA m^−2^) was calculated from *E*, *R* and the anode area (20 cm^2^). Two MFCs were set up, and they were operated at hydraulic residence times (HRTs) of 5 and 40 days (termed HRT5 and HRT40, respectively). The HRT was controlled by replacing a certain amount of electrolyte to the fresh ADE every two days (32 mL for HRT5 and 4 mL for HRT40). Polarization analyses were conducted using a potentiostat (HZ-5000, Meiden Hokuto, Tokyo, Japan), as described previously [20], when the cell voltage was close to the highest value in one cycle. A power density (*P*, mW m^−2^) was calculated from *E*, *J* and the anode area. The maximum power density (*P*_max_, mW m^−2^) was the peak in a *J* vs. *P* curve.

After the operation of MFC, bacteria grown in the electrolyte (planktonic microbes, PM) and those on the anode (anode biofilm, AB) were analyzed by metabarcoding of PCR-amplified 16S rRNA gene fragments, as described previously [20]. Briefly, the V3/V4 region of bacterial 16S rRNA genes was amplified by PCR using universal primers, and amplicons were subjected to sequencing using Miseq sequencer (Illumina, San Diego, CA, USA). The sequences obtained were clustered into operational taxonomic units (OTUs) with 97% similarity using QIIME 2 [21] and taxonomically classified by aligning these with sequences in the Greengene database [22]. The sequences generated in the metabarcoding analysis were deposited in the DDBJ Sequence Read Archive database under accession number DRA017553.

### 2.2. Isolation of EAB

The following procedures were conducted in an anaerobic chamber (Bactron, Sheldon Manufacturing, Cornelius, OR, USA). Small pieces of anodes in MFCs were cut, soaked in a DSM826 medium and intensely vortexed for suspending anode-adhering microbes into the medium. This medium contains (per liter) NH_4_Cl 1.50 g, Na_2_HPO_4_ 0.60 g, KCl 0.10 g, NaHCO_3_ 2.50 g, sodium acetate 0.82 g, sodium fumarate 8.00 g, modified Wolin’s mineral solution 10.00 mL and Wolin’s vitamin solution (10x) 1.00 mL (Deustsche Sammlung von Mikroorganismen und Zellkulturen) and has been used for routine cultivation of bacteria affiliated with the genus *Geobacter* [17,23]. Microbial suspensions were serially diluted and spread onto agar plates containing a DSM826 medium. The plates were put in anaerobic bags (Anaeropack; Mitsubishi Gas Chemical Company, Tokyo, Japan) and incubated at 30 °C. Red colonies formed on the plates were picked using needles and streaked on fresh DSM826 plates. This procedure was repeated several times, until only colonies with uniform morphology were formed.

For examining their growth in liquid cultures, isolates were cultivated in test tubes containing DSM826, and growth was monitored by measuring optical density at 600 nm (OD_600_) using a mini photo 518R photometer (Taitec, Tokyo, Japan). Liquid cultures were preserved at −80 °C in the presence of 20% dimethyl sulfoxide.

Partial fragments of 16S rRNA genes were PCR-amplified using primers 515f and 1492r [24], and amplicons were sequenced by the standard procedure. Sequences thus obtained were used to analyze their phylogeny using the BLAST program in the NCBI database [25], and phylogenetic trees were constructed by the neighbor-joining method using the MEGA software ver. 11 [26]. The nucleotide sequences for the strains ADMFC1, ADMFC2 and ADMFC3 determined in this study have been deposited in the GSDB, DDBJ, EMBL, and NCBI nucleotide sequence databases under accession nos. LC790428, LC790429 and LC790430, respectively.

### 2.3. Electron-Donor Tests

The basal medium was DSM 826, from which acetate was omitted. In this medium, fumarate could serve as the electron acceptor. One of the electron donors listed in Table 1 was added to the medium at a concentration of 10 mM. The medium was put in a test tube equipped with a screw cap and a butyl rubber stopper, and the head space was filled with high-purity nitrogen gas (99.999% in purity). Growth was checked in triplicate or more replicates under the anaerobic condition at 30 °C by measuring OD_600_ using the photometer.

### 2.4. Electrochemical Characterization

Single-chamber three-electrode electrochemical cells (ECs, 95 mL in capacity) were used to evaluate current generation in pure cultures. An EC was equipped with a working electrode (WE; graphite felt, 2 cm^2^ in area, 3 mm in thickness; GF-20-5F, Nippon carbon, Tokyo, Japan), an Ag/AgCl reference electrode (+0.2 V vs. standard hydrogen electrode, SHE; RE-T7A, EC Frontier, Kyoto, Japan), and a counter electrode (CE, platinum wire; 15 cm × 0.3 mm; Nilaco, Tokyo, Japan). The EC was filled with 65 mL of fumarate-free DSM 826 medium (containing 10 mM acetate as the electron donor), and after being purged with high-purity nitrogen gas and supplemented with L-cysteine (5 mM), it was inoculated with a bacterial culture pre-grown in the DSM826 medium. The initial OD_600_ was 0.02. An EC was connected to a potentiostat (HA151-B, Meiden Hokuto, Tokyo, Japan), and, after the WE potential was poised at +0.2 V (vs. Ag/AgCl), the current (*I*, mA) was measured and recorded. The current density (*J*, mA cm^−2^) was calculated based on the projected area of WE. The measurement was conducted several times for each strain and reproducibility was checked.

During the chronoamperometric measurement, the WE was subjected to cyclic voltammetry (CV) using a potentiostat (HZ-5000). The WE potential (*P*, V) was reciprocally swept (more than three times) between −0.8 V and +0.8 V vs. SHE at a scan rate of 1 mV s^−1^, and the current (*I*, mA) was recorded.

### 2.5. Cytochrome c (cyt-c) Content

The cyt-c content of bacterial cells was determined from the absorption spectrum of a cell-free extract and protein content, according to methods described elsewhere [27] with modifications. Bacteria strains were cultivated anaerobically in DSM826 medium, and cells were collected by centrifugation at 10,000 ×*g* for 15 min. After the cell pellet was washed with TE buffer, the cells were suspended in TE buffer containing lysozyme (1 mg mL^−1^, Fuji Film Wako, Tokyo, Japan). The suspension was treated with a homogenizer and incubated for 1.5 h at 37 °C. Triton X-100 (1% w/v, Fuji Film Wako, Tokyo, Japan) was added to the suspension and further incubated for 10 min at room temperature. It was centrifuged at 10,000 ×*g* for 15 min., and the supernatant was recovered. An absorption spectrum of the supernatant was measured in a range from 700 to 350 nm using a spectrophotometer (UH5300, Hitachi, Tokyo, Japan). In this measurement, cyt-c was fully oxidized. For measuring an absorption spectrum for the reduced form of cyt-c, sodium hydrosulfite was added at a concentration of 68 mM, and a differential spectrum (reduced minus oxidized) was obtained. The protein content (mg) of the cell suspension was determined using a Micro BCA protein assay kit (Thermo Fisher Scientific Japan, Tokyo, Japan), as described previously [28]. For comparison, the cell-free extract of *Geobacter sulfurreducens* PCA [23] was also measured. The measurement was conducted several times for each strain and reproducibility was checked.

## 3. Results and Discussion

### 3.1. Enrichment of EAB

As the first step for isolating EAB from ADE, the EAB were enriched in two MFCs (MFC5 and MFC40) operated at different HRTs. As shown in Figure 1A, the voltage outputs from these MFCs were stable during the 40-day operation. From the polarization analyses conducted on day 38, the *P*_max_ values for MFC5 and MFC40 were estimated to be 450 mW m^−2^ and 230 mW m^−2^, respectively, and these values are similar to those reported previously [20]. The high *P*_max_ value of MFC5 (compared to that of MFC40) can be attributed to the high organics-loading rate of this reactor, which provided more organic substrates to the EAB in MFC5 than to those in MFC40.

In order to detect the bacteria that occurred in MFCs, the PM and AB fractions were subjected to metabarcoding analysis (Figure 1B). Since no information is available on the amounts of microbes that occurred in the PM and AB fractions, only the relative abundances of major bacterial groups in each fraction can be inferred from the metabarcoding data. On the whole, this analysis abundantly detected families within the phyla *Bacillota* and *Bacteroidota* that have also been frequently detected in anaerobic digesters in previous studies [29,30]. In addition to these, families affiliated with the phylum *Thermodesulfobacteriota*, such as *Pelobacteraceae* and *Desulfuromonadaceae*, were also detected abundantly in the AB fractions. These families are known to include EAB [9]. The relative abundances of some bacterial families, including *Anaerolineae* SJA-15 and *Ruminococcaceae*, in the AB fraction of MFC5 were higher than those in the PM fraction of MFC5, while *Pelobacteraceae* and *Bacteroidales* were abundant in the AB fraction of MFC40. The bacterial groups specifically detected in the AB fractions are of interest, since such bacteria are considered to include EAB involved in current generation in MFCs [19,31].

### 3.2. Isolation of Bacterial Strains

Anode-associated microbes were grown on DSM 826 agar plates, and reddish colonies were purified. Several weeks were needed for the appearance of visible red colonies on the plates. Repeated cultivation on the plates (more than three times) resulted in the occurrence of colonies with uniform morphology on single plates (Figure 2), and we decided to analyze three strains in subsequent experiments. Among these, one strain, named ADMFC1, was isolated from the MFC5 anode, while two strains, named ADMFC2 and ADMFC3, were isolated from the MFC40 anode. The colony morphologies of the three strains on DSM826 agar plates are shown in Figure 2. As shown in these photos, the three strains form reddish colonies. It is also shown that, compared to the strains ADMFC1 and ADMFC2, ADMFC3 exhibits a dark red color.

### 3.3. Phylogenetic Characteristics of Isolates

In order to determine the phylogenetic characteristics of the isolates, nucleotide sequences of partial 16S rRNA gene fragments were subjected to the BLAST search (Table 2). It is shown that they are phylogenetically diverse and belong to different phyla. Among them, sequence homology between ADMFC1 and the closest relative in the Blast search (*Desulfosporosinus acididurans*) is 88.8%, suggesting the possibility that this strain represents a novel genus. The closest relative of ADMFC2 is *Sulfurospirillum alkalitolerans* (93.8%), suggesting the possibility that this strain also represents a novel genus. In addition, the closest relative of ADMFC3 is *Geovibrio thiophilus* (97.5%), suggesting the possibility that this strain represents a novel species in the genus *Geovibrio*. Among them, the genera *Sulfurospirillum* and *Geovibrio* have been described to include EAB [9], while their electrochemical properties have not yet been examined in detail. It is interesting that the three strains isolated in the present study are affiliated with different phyla. It is therefore conceivable that anaerobic digesters harbor diverse EAB.

In order to gain information as to possible phylogenetic relationships between these isolates and their close relatives, phylogenetic trees were constructed using partial 16S rRNA gene sequences (Figure 3). ADMFC1 seems to form a distinct lineage that is separated from the known genera in the order *Eubacterales* (Figure 3A) and may represent a novel genus. According to Figure 3B, the strain ADMFC2 seems to be included in the genus *Sulfurospirillum*. However, we suggest the possibility that this strain could also represent a novel genus for the following reasons. First, the genus *Sulfurospirillum* may be divided into two genera, since homologies of 16S rRNA gene sequences (almost full-length sequences in the databases) between the two groups of *Sulfurospirillum*, represented by *Sulfurospirillum multivorans* and *Sulfurospirillum alkalitolerans*, are less than 92%. Second, the sequence homologies between ADMFC2 and the *Sulfurospirillum* strains are less than 94%. ADMFC3 is considered to be affiliated with the genus *Geovibrio*, while it may not belong to the known species (Figure 3C). For species identification, genome comparisons would be necessary [32]. Further studies are therefore necessary for the taxonomic identification of the three isolates, in which multiple analyses will be conducted, including phylogenetic analyses using complete sequences of 16S rRNA genes, genomic taxonomy, and physiological analyses.

### 3.4. Electrochemical Characteristics of Isolates

Before investigating if the three isolates are EAB, we examined organic substrates that support growth of the three isolates. The results are summarized in Table 1. In this table, the cultures in which bacteria grew fermentatively were also indicated as positive. As shown in this table, ADMFC3 is able to utilize the largest number of organic substrates among the three strains. It is also shown in Table 1 that these strains exhibit different preferences for organic substrates.

In order to examine whether the three isolated strains are EAB, ECs were inoculated with these strains, and chronoamperometric measurements were conducted at the WE potential of +0.4 V (vs. SHE) in the presence of acetate (10 mM) as the electron donor (Figure 4). This potential was chosen according to previous work [16]. Since hydrogen may be evolved at the CE, hydrogen may also serve as an electron donor. As shown in these graphs, all the three strains generated substantial amounts of currents, demonstrating that they have electrochemical activities. Among them, the strain ADMFC1 generated the highest current density. This level of current density is considered to be higher than that attained by *Shewanella oneidensis* MR-1 [33] and similar to that attained by *Geobacter sulfurreducens* PCA [34]. In addition, studies have identified a thermophilic bacterial strain, *Thermincola* sp. strain JR, affiliated with the phylum *Bacillota*, to be an active EAB [27]. Although direct comparison between the strains JR and ADMFC1 in a same system has not been conducted, we assume that these EAB strains would have similar levels of electrochemical activities.

The strain ADMFC2 is considered to be the first EAB identified in the family *Sulfurospirillaceae*. In the phylum *Campylobacterota*, an EAB affiliated with the genus *Arcobacter* has been reported [35]. ADMFC2 is therefore considered to be the second EAB identified in the phylum *Campylobacterota*. Concerning ADMFC3, the phylum *Deferribacterota* is known to include iron-reducing bacteria, such as the *Deferribacter* and *Geovibrio* strains [36,37]. In addition, studies have detected these bacteria in thermophilic MFCs [38,39]. To our knowledge, however, no study has described their electrochemical activities, including current generation, examined in pure cultures. The data reported for ADMFC3 would therefore be of value for confirming that *Deferribacterota* includes EAB.

During the chronoamperometric measurements, the CV patterns of the three strains (in the presence of acetate as the electron donor) were compared (Figure 5). It is presented in this figure that the three strains exhibit different CV patterns. ADMFC1 generated anodic current from the lowest potential (around −0.4 V vs. SHE), and increases in current were observed in two potential regions (centered at −0.2 V and +0.3 V). This suggests that two different redox components are involved in the electrochemical activity of ADMFC1. It has been reported that *G. sulfurreducens* biofilm exhibits a major redox peak centered at −0.2 V vs. SHE in CV analyses, and the peak is ascribable to its cytochrome-based EET pathway [40]. On the other hand, single increases in current were observed for ADMFC2 and ADMFC3 (centered at +0.3 V and +0.5 V, respectively). It is interesting that the three EAB isolated from the same habitat exhibit different electrochemical characteristics. Future studies will therefore address how these bacteria utilize different electrochemical properties for their survival in the anaerobic digester.

### 3.5. Cyt-c Contents

It has been known that the cells of some EAB, including those affiliated with *Geobacter*, *Shewanella* and *Thermincola*, are reddish, since they express substantial amounts of cyt-c as major components of their EET pathways [9]. The three EAB isolated in the present work also form reddish colonies on agar plates (Figure 2). We therefore assumed that the three EAB also contain substantial amounts of cyt-c in their cells and examined this possibility in the spectrometric observation. Figure 6 shows differential spectra (reduced minus oxidized forms) for cell-free extracts of the three EAB, and these spectra are compared to that of *G. sulfurreducens* PCA. In Figure 6, typical differential spectra for cyt-c are observed for all the strains tested, indicating that these bacteria contain substantial amounts of cyt-c in their cells. It is also shown that the cells of ADMFC2 and ADMFC3 contain larger amounts of cyt-c than PCA, while the amount of cyt-c in ADMFC1 is relatively small. It is interesting that the amounts of cyt-c are not correlated with the current densities attained by the three strains (Figure 4). Several possible explanations are possible for this result. First, current generation by bacteria is dependent on multiple cellular activities, including those for catabolism, EET and biofilm formation [33]. Second, some bacteria, including gram-positive bacteria, exploit cyt-c-independent EET pathways, e.g., those involving flavins [9]. In addition, it has been known that *G. sulfurreducens* cells contain a variety of cyt-c, only some of which are involved in EET [23]. It would be of interest to determine the genome sequences of the three isolates and identify the cyt-c genes that are up-regulated under current generation.

## 4. Conclusions

Three EAB were isolated from the anaerobic digester treating food wastes after enrichment in MFCs. These isolates are phylogenetically diverse and affiliated with three different phyla, including *Bacillota*, *Campylobacterota* and *Deferribacterota*. It is interesting that the EAB affiliated with these phyla were primarily isolated ahead of well-studied EAB, such as those affiliated with the genus *Geobacter*, even though the medium used for the routine cultivation of *Geobacter* bacteria was used for the isolation of these strains. In addition, these EAB exhibit different electrochemical characteristics, as assessed by the CV analyses, and contain substantial amounts of cyt-c in their cells. These results suggest that anaerobic digesters harbor diverse EAB, and it would be of interest to gain information as to how they thrive in anaerobic digesters. Previous studies have shown electrochemical interactions between EAB and methanogens [41,42], suggesting the possibility that EAB assist with methanogenesis. In addition, electron exchanges between different species of EAB (electric syntrophy) may also be common in anaerobic habitats [43,44]. In order to examine these possibilities, defined mixed-culture experiments would be useful. On the other hand, the present work expands our knowledge on the diversity of EAB, implying that more diverse EAB are present in nature than previously known. The isolation and characterization of novel EAB from natural habitats would be of value not only for fundamental science but also for their application in the development and improvement of MESs, such as their use as bioaugmentation agents.

## Figures and Tables

**Figure 1 microorganisms-12-01645-f001:**
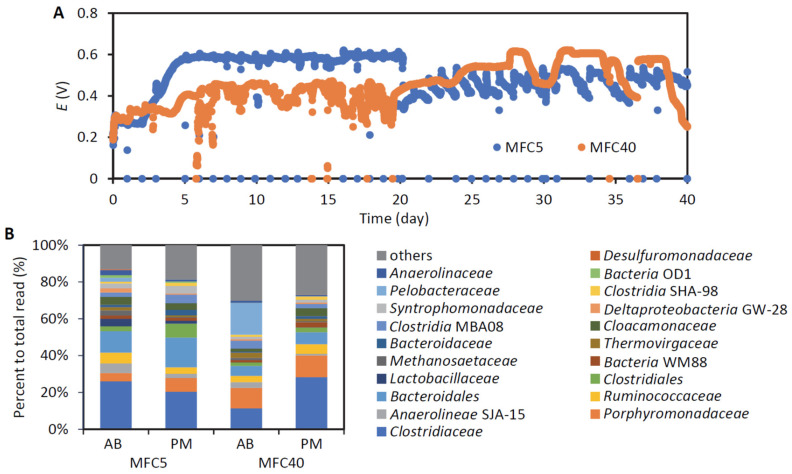
Enrichment of EAB in MFCs. (**A**) Time courses of cell voltages. The external resistor of MFC5 was changed to 500 Ω on day 21. (**B**) Metabarcoding of bacteria in anode biofilm (**A**,**B**) and planktonic microbes (PM) in MFCs. OTUs were assigned to family-level taxonomic groups, and major families were compared. When an OTU is not affiliated with a known family, it is assigned to a taxonomic group of a higher rank.

**Figure 2 microorganisms-12-01645-f002:**
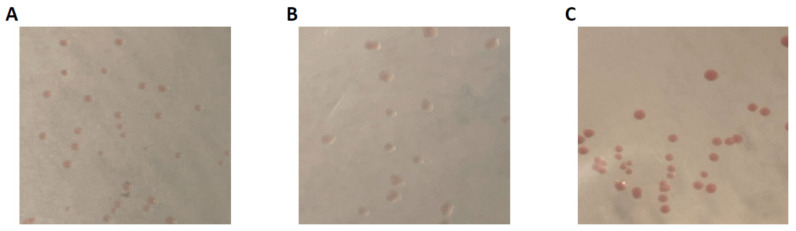
Colonies of strains ADMFC1 (**A**), ADMFC2 (**B**) and ADMFC3 (**C**) formed on DSM826 ager plates.

**Figure 3 microorganisms-12-01645-f003:**
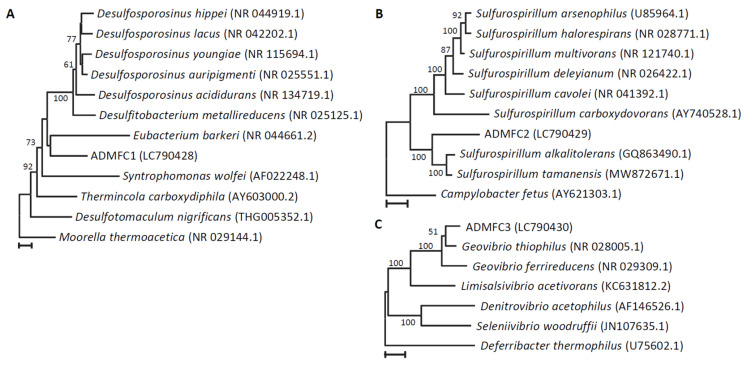
Phylogenetic trees based on 16S rRNA gene sequences showing relationships between the isolated strains (**A**, ADMFC1; **B**, ADMFC2; **C**, ADMFC3) and close relatives. Accession numbers for the sequences retrieved from the databases are given in parentheses. The numbers at branch nodes are bootstrap values (per 100 trials); only values greater than 50 are shown. The scale bars indicate 0.02 substitution per sites.

**Figure 4 microorganisms-12-01645-f004:**
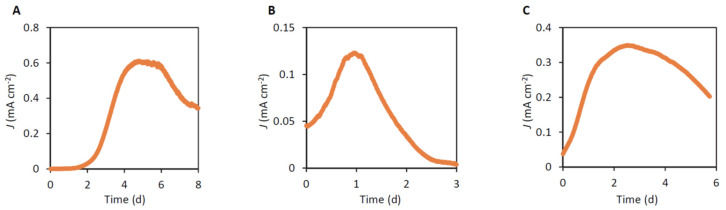
Current generation by strains ADMFC1 (**A**), ADMFC2 (**B**) and ADMFC3 (**C**) in BESs with 10 mM acetate as the electron donor. Anode potentials were poised at +0.4 V vs. SHE.

**Figure 5 microorganisms-12-01645-f005:**
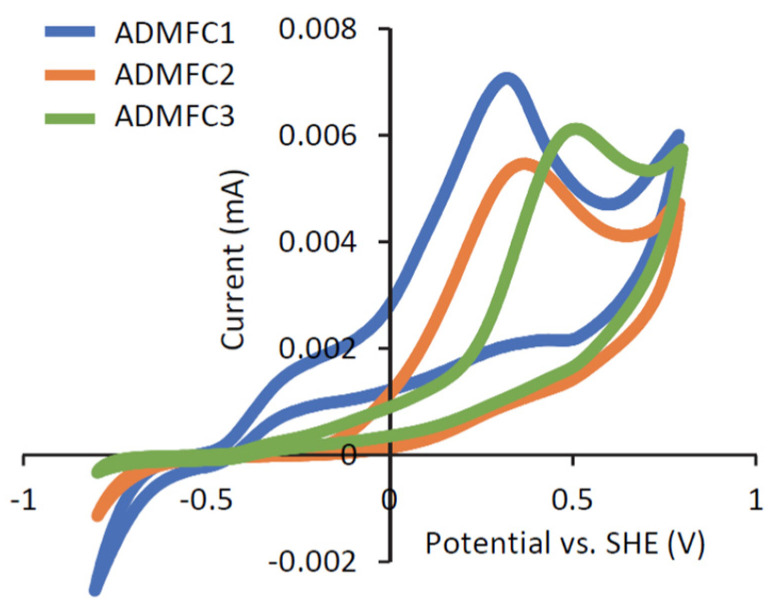
Electrochemical properties of strains ADMFC1, ADMFC2 and ADMFC3, as assessed by comparative CV analyses in the presence of acetate as the electron donor. Curves obtained in the second cycles are shown.

**Figure 6 microorganisms-12-01645-f006:**
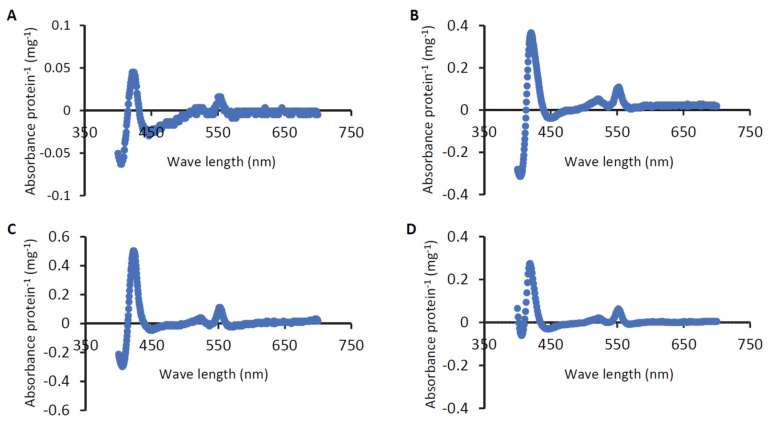
Differential spectra (reduced minus oxidized forms) for detergent-solubilized cell-free extracts of strains ADMFC1 (**A**), ADMFC2 (**B**) and ADMFC3 (**C**). For comparison, a spectrum for *G. sulfurreducens* is also shown in panel **D**.

**Table 1 microorganisms-12-01645-t001:** Organic substrates that support the growth of the three isolates.

Substrate	ADMFC1	ADMFC2	ADMFC3
Glucose	−	−	+
Ethanol	+	−	+
Lactate	−	+	+
Pyruvate	−	+	+
Acetate	+	+	+
Formate	+	−	+
Hydrogen	+	−	−

**Table 2 microorganisms-12-01645-t002:** Phylogenetic characteristics of the three isolates, as assessed by the BLAST search of PCR-amplified partial 16S rRNA gene fragments.

Strain	Closest Relative (% Identity)	Phylum
ADMFC1	*Desulfosporosinus acididurans* (88.8%)	*Bacillota*
ADMFC2	*Sulfurospirillum alkalitolerans* (93.8%)	*Campylobacterota*
ADMFC3	*Geovibrio thiophilus* (97.5%)	*Deferribacterota*

## Data Availability

The data of this study are available from the corresponding author upon reasonable request.

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
