# Peer review of "Isolation of Electrochemically Active Bacteria from an Anaerobic Digester Treating Food Waste and Their Characterization"

_microorganisms, 2024, doi:10.3390/microorganisms12081645_

Round 1

Reviewer 1 Report

Comments and Suggestions for Authors

Yoshizu et al. used microbial fuel cells (MFCs) to enrich electroactive biofilm from anaerobic-digester effluent. Using several experimental tools, they were able to isolate three phylogenetically diverse strains affiliated with the phyla Bacillota, Campylobacterota, and Deferribacterota. Electrochemical examination reveals the high efficiency of all isolated strains to generate electric power in MFCs. This study demonstrates that anaerobic-digester effluent might be a promising stream that harbors diverse electroactive biofilm with a biotechnological application in generating electricity from wastewater. Generally, the article is well written and fits within the scope of the journal. I have a few comments before its acceptance as follows:

(1) Why did MFC5 exhibit high energy production compared to MFC40?

(2) The authors constructed the polarization plot at two different conditions. After day 38, it is expected to have low organic matter in MFC40 compared to MFC5. Did they perform polarization analyses right after replacing the growth medium for both MFCs?

(3) Did the authors perform EIS of isolated electroactive strains?

Comments on the Quality of English Language

There are a few grammatical typos, which should be corrected.

Author Response

(1) Why did MFC5 exhibit high energy production compared to MFC40?

We consider that this was because more organic substrates were supplied to EAB in MFC5 than those in MFC40. This notion is added in the revised manuscript (L177 to L178 in the revised MS).

(2) The authors constructed the polarization plot at two different conditions. After day 38, it is expected to have low organic matter in MFC40 compared to MFC5. Did they perform polarization analyses right after replacing the growth medium for both MFCs?

We conducted polarization analyses right after ADE was added to MFCs and when cell voltages were close to the highest value in one cycle (L83).

(3) Did the authors perform EIS of isolated electroactive strains?

We did not conduct EIS and have no data. We consider EIS has not been used commonly for comparing electrochemical properties of EAB.

Reviewer 2 Report

Comments and Suggestions for Authors

The isolation of electrochemically active bacteria from an anaerobic

digester treating food wastes and their characterization was shown in this paper.

In this research the single-chamber MFC was used. The wastewater from anaerobic-digester effluent as a source of organics and microbes was used. During work the red colonies were selected from those grown on the agar plates, resulting in the isolation of three phylogenetically diverse strains affiliated with the phyla Bacillota, Campylobacterota and Deferribacterota. It was noted that all selected strains are capable of current generation, but they exhibit different electrochemical properties as assessed by cyclic voltammetry. Analyses of cell-free extracts conducted in this work indicate a high presence of cytochromes in their cells, suggesting a role in current generation. It was rightly noted that the results suggest that anaerobic digesters contain diverse electroactive bacteria (EAB), and it would be valuable to study their ecological niches in anaerobic digestion.

The work is well described and redable.

The methodology is well thought out.

The experiment was well planned and carried out.

Figures are legible, but their resolution could be improved.

The references are well matched to the topic of the work.

The work presents the next step in EAB research. EAB research is extremely important in developing acceptably efficient MFCs. Therefore, the work this is important from the point of view of analyzing the development of a suitable electrode biofilm for MFC. The work is both a source of new information and will certainly inspire other researchers to undertake further work on efficient MFCs.

Some suggestions follow:

- It would be worth mentioning any difficulties encountered during the preparation or execution of the experiment. This type of information could serve as a guide for other researchers, either to avoid the same pitfalls or, conversely, to encourage deeper exploration in challenging areas.

- The conclusions are well-written, but it would be beneficial to emphasize your own scientific contributions more. This will enhance both the substantive value and the appeal to readers

Good work!

Author Response

The work is well described and redable.

The methodology is well thought out.

The experiment was well planned and carried out.

Figures are legible, but their resolution could be improved. 

Response: The resolution of all figures was improved.

The references are well matched to the topic of the work.

The work presents the next step in EAB research. EAB research is extremely important in developing acceptably efficient MFCs. Therefore, the work this is important from the point of view of analyzing the development of a suitable electrode biofilm for MFC. The work is both a source of new information and will certainly inspire other researchers to undertake further work on efficient MFCs.

Some suggestions follow:

It would be worth mentioning any difficulties encountered during the preparation or execution of the experiment. This type of information could serve as a guide for other researchers, either to avoid the same pitfalls or, conversely, to encourage deeper exploration in challenging areas.

Response: We need substantial amounts of time for isolation of EAB. In oder for readers to know much time is necessary for isolation of EAB, we added this information (L203 to L206 in the revised MS).

- The conclusions are well-written, but it would be beneficial to emphasize your own scientific contributions more. This will enhance both the substantive value and the appeal to readers

Response:  A sentence is added in the conclusion section to appeal the interesting point of this work (L325 to L328).

Reviewer 3 Report

Comments and Suggestions for Authors

The presented paper is of interest for the development and application of biofuel cells based on the bacteria from an anaerobic digester for energy production. In addition, the paper presents some new data on the isolation and analysis of microorganisms of the anaerobic microbial community. In general this article is in line with the theme of the journal Microorganisms. However, there are a few important points that require revision before the manuscript can be accepted for publication:

1. The manuscript is titled "Isolation of electrochemically active bacteria from an anaerobic digester treating food wastes and their characterization", but the characteristics of the new bacterial isolates are not well presented. It would be worthwhile to include biochemical assessments of the behavior of the new isolates in order to better understand their specific metabolic processes.

2. The authors of the paper should elaborate on the novelty of their manuscript. The use of activated sludge microorganisms in a microbial fuel cell is not a new concept, and approaches to the isolation and characterization of individual bacteria are generally well established.

3. In string 75, the authors mention that they used the resistance of R, Ω; initially 2,000 Ω. It would be helpful to explain how this specific value of external resistance was chosen.

4. The authors should provide a сircuit showing the formation and path of the electron flow within the considered MFC model. This would help to better understand the processes occurring inside a single chamber MFC.

5. It would be helpful to provide an explanation how a potential of WE +0.4 V vs SHE was chosen for the chronoamperometric measurements (str. 254).

6. Considering on Fig. 4, it appears that each microorganism exhibits a distinct characteristic of the generated current. The authors could provide more detailed discussion of the dependence of the generated current on time. It may also be worthwhile to compare these data with growth curves of isolated microorganisms.

Comments on the Quality of English Language

English language require some polishing.

Author Response

1. The manuscript is titled "Isolation of electrochemically active bacteria from an anaerobic digester treating food wastes and their characterization", but the characteristics of the new bacterial isolates are not well presented. It would be worthwhile to include biochemical assessments of the behavior of the new isolates in order to better understand their specific metabolic processes.

Response: We have shown metabolic substrates that support growth of these isolates. In addition, electrochemical characteristics, phylogenetic characteristics and the presence of large amounts of cytochromes in their cells are shown. All these are included in "characterization".

2. The authors of the paper should elaborate on the novelty of their manuscript. The use of activated sludge microorganisms in a microbial fuel cell is not a new concept, and approaches to the isolation and characterization of individual bacteria are generally well established.

Response: We did not use activated sludge as the source of EAB. What we used was effluent from an anaerobic digester. Isolation of microbes is the basis of microbiology, and substantial amounts of microbes, including bacteria, have been isolated and characterized. Onthe other hand, limted studies have described the isolation of EAB, and, to our knowledge, no study has isolated EAB from anaerobic digesters. This would be the novel point of this work as described in the introduction.

3. In string 75, the authors mention that they used the resistance of R, Ω; initially 2,000 Ω. It would be helpful to explain how this specific value of external resistance was chosen.

Response: This resister was selected based on our previous work (Yoshizu et al. 2023), in which high power output was obtained in MFCs with the same configuration. This is additionally described in the revised manuscript (L76 to L77 in the revised MS).

4. The authors should provide a сircuit showing the formation and path of the electron flow within the considered MFC model. This would help to better understand the processes occurring inside a single chamber MFC.

Response: You can find the structure and electron flow in reference 3. Reference 3 was changed from that in the original manuscript, because the current one would be better suited to this manuscript. This reference is cited in line 70.

5. It would be helpful to provide an explanation how a potential of WE +0.4 V vs SHE was chosen for the chronoamperometric measurements (str. 254).

Responses: This potential has been used in previous studies for examining current generation by EAB. A sentence is added to explain this with a reference paper (L266 in the revised MS).

6. Considering on Fig. 4, it appears that each microorganism exhibits a distinct characteristic of the generated current. The authors could provide more detailed discussion of the dependence of the generated current on time. It may also be worthwhile to compare these data with growth curves of isolated microorganisms.

Response: This would be an important point. However, since most EAB grow in anode biofilms when generating current, monitoring of growth during current generation is very difficult. In the present work, we therefore did not conduct growth measurement, and we have no data.

Reviewer 4 Report

Comments and Suggestions for Authors

Authors here present the isolation of electrochemically active bacteria from anaerobic digester claiming it as the fist of its kind in the literature. There are several other studies reported on EAB (both cathode and anode) from different sources, please do the literature work and consider presenting the claim. The methodology is incomplete with microbial, genomics and electrochemical information which has to be included for an independent researcher to reproduce this work. Please also show the replicates and statistics with the data. At present conditions it does not appears to be reliable study.

The title says EAB isolated from anaerobic digesoer but as the methodology goes it says the cultures were operated in MFC mode for days icoculated with ADE effluent. This will certainy cause the microbial community to switch favoring electroactive bacteria to survive/dominate replinishing non-electroactive in the environment. Please refer to microbial ecology papers for better understanding of microbial community dynamics with change in different environments.

Authors are advised to carefully check spellings and terminologies throughout manuscript, eg: Vortexed not voltexED (L97)

Table2, authors should mention the output and discuss their observation on different carbon sources with the isolates. it is obvious that bacterium can utilize and metabolize more than one carbon source. Use of glucose could result in fermentation rather what claimed in the study. Did authors check the recovery/final concentrations of these copounds after every cycles? Did HPLC showed varied concentrations of these compounds over time against the no inocula control?

Electrochemical part is incomplete. Please provide reason or cite article on set conditions and make it clear for readers to understand. Authors are encouraged to justify the use of chronoamperommetry, the methodology mentioned is not correct which wuthors have mixed with cyclic voltametry. CA is a technique where you apply certain potential and measure current, what was the applied potential, conditions, electrode system and time? For CV, please mention the electrolyte condition, electrode system, scane rate, potential range and number of cycle.

Were there any replicates for this studies? I do not see data on controlled MFC/experiments. How many replicates were used ?

The genomic part is unclear and incomplete. Did authors performed the sequence analysis using bioinformatics tools? please mention that. Was it the whole genome or just the 16S rRNA sequence which claimed to be identified species. to claim the novel species it is always encouaged to consider whole genome sequence, assemble the contigs checking the contamination, analyze and look for phlylogenetics.

Was that isolate from a single MFC? 

Did authors check for microbial communities in the initial and final stage of MFC? 

which among the particluar MFC replicate this isolate obtained from? did other replicate MFCs showed presence of this species too?

It is surprising to see that despite no sulfur in the media (other than amino acid cyctein) a sulfur reducing bacterium was dominating one and isolated. can authors explain this ?

Please avoid irrlevant referances and stick to articles related to this work a lot of losely related and old articles can be avoided inseated include recent and core articles related to this work.

Comments on the Quality of English Language

Fine but minor corrections are required.

Author Response

Authors here present the isolation of electrochemically active bacteria from anaerobic digester claiming it as the fist of its kind in the literature. There are several other studies reported on EAB (both cathode and anode) from different sources, please do the literature work and consider presenting the claim. The methodology is incomplete with microbial, genomics and electrochemical information which has to be included for an independent researcher to reproduce this work. Please also show the replicates and statistics with the data. At present conditions it does not appears to be reliable study.

Response: Thoughout the work, we checked reproducibility of experimental results and this is described in the revised manuscript where appropriate. Methods section was partly revised.

The title says EAB isolated from anaerobic digesoer but as the methodology goes it says the cultures were operated in MFC mode for days icoculated with ADE effluent. This will certainy cause the microbial community to switch favoring electroactive bacteria to survive/dominate replinishing non-electroactive in the environment. Please refer to microbial ecology papers for better understanding of microbial community dynamics with change in different environments.

Response: The original source of microbes was an anaerobic digester, where the isolated strain must thrive. We therefore consider the strains were isolated from the anaerobic digester.

Authors are advised to carefully check spellings and terminologies throughout manuscript, eg: Vortexed not voltexED (L97)

Response: Spellings were checked thoughout the manuscript.

Table2, authors should mention the output and discuss their observation on different carbon sources with the isolates. it is obvious that bacterium can utilize and metabolize more than one carbon source. Use of glucose could result in fermentation rather what claimed in the study. Did authors check the recovery/final concentrations of these copounds after every cycles? Did HPLC showed varied concentrations of these compounds over time against the no inocula control?

Response: We did not analyze concentrations of growth substrates and metabolites. We agree that they may have grown fermentatively, and sentences and Table 1 were revised to reflect this possibility (L252 to L257 in the revised manuscript). 

Electrochemical part is incomplete. Please provide reason or cite article on set conditions and make it clear for readers to understand. Authors are encouraged to justify the use of chronoamperommetry, the methodology mentioned is not correct which wuthors have mixed with cyclic voltametry. CA is a technique where you apply certain potential and measure current, what was the applied potential, conditions, electrode system and time? For CV, please mention the electrolyte condition, electrode system, scane rate, potential range and number of cycle.

Response: Most of the above-mentioned information have already been described in the method section except for cycle number of CV. This information is added in the revised manuscript (L148). During the current measurement at +0.4 V vs. SHE, the potentiostat connected to the bioelectrochemical cell was changed to another one that can be used for CV analyses.

Were there any replicates for this studies? I do not see data on controlled MFC/experiments. How many replicates were used ?

Response: Experiments for characterization of isolates were conducted repeatedly to confirm reproducibility. This information is added in the revised manuscript where appropriate (including L127, L144, and L167).

The genomic part is unclear and incomplete. Did authors performed the sequence analysis using bioinformatics tools? please mention that. Was it the whole genome or just the 16S rRNA sequence which claimed to be identified species. to claim the novel species it is always encouaged to consider whole genome sequence, assemble the contigs checking the contamination, analyze and look for phlylogenetics.

Response: This study did not conduct any genomic analysis. What we did was phylogenetic analyses of the isolated strains based on 16S rRNA gene sequences, and methods for this analysis are described at the end of section 2.2.

Was that isolate from a single MFC? 

Response: Among the three isolates, one strain was isolated from MFC5, while the other two were isolated from MFC40 as described (L206 to L208).

Did authors check for microbial communities in the initial and final stage of MFC? 

Response: We only analyzed microbial communities at the final stage of MFCs. 

which among the particluar MFC replicate this isolate obtained from? did other replicate MFCs showed presence of this species too?

Responses: Among the three isolates, one strain was isolated from MFC5, while the other two were isolated from MFC40 as described (L206 to L208). We neither detected nor isolated these strains from another MFC, while, even so, we could not say that these were not present in another MFC.

It is surprising to see that despite no sulfur in the media (other than amino acid cyctein) a sulfur reducing bacterium was dominating one and isolated. can authors explain this ?

> Sulfurospirillum bacteria are know to be sulfur oxidizers/reducers, while they may also grow on different types of energy metabolism. ADMFC2 is unique in this ability to utilize electrodes as electron acceptors.

Please avoid irrlevant referances and stick to articles related to this work a lot of losely related and old articles can be avoided inseated include recent and core articles related to this work.

> We consider all references cited are necessary for the context of this manuscript.

Round 2

Reviewer 4 Report

Comments and Suggestions for Authors

Authors have failed to address the comments raised and implement the suggestions. The revised version appears as the added sentances were made to answer the reviewer's comments, particulary with spelling mistakes (resistor not resister). This is the quickest revision I have received for an article.

The interest and important authors have shown on this work is very disappointing and not safe for the scientific publication. Ignoring to provide the crucial data on the presented work speaks volumes of it.

I strongly recommend REJECT of this article for the publication at current state. 

Comments on the Quality of English Language

Minor edits are required.